# Are the Effects of Malnutrition on the Gut Microbiota–Brain Axis the Core Pathologies of Anorexia Nervosa?

**DOI:** 10.3390/microorganisms10081486

**Published:** 2022-07-24

**Authors:** Stein Frostad

**Affiliations:** Division of Psychiatry, Haukeland University Hospital, 5021 Bergen, Norway; stein.frostad@helse-bergen.no

**Keywords:** anorexia nervosa, anxiety, gut microbiota–brain axis

## Abstract

Anorexia nervosa (AN) is a disabling, costly, and potentially deadly illness. Treatment failure and relapse after treatment are common. Several studies have indicated the involvement of the gut microbiota–brain (GMB) axis. This narrative review hypothesizes that AN is driven by malnutrition-induced alterations in the GMB axis in susceptible individuals. According to this hypothesis, initial weight loss can voluntarily occur through dieting or be caused by somatic or psychiatric diseases. Malnutrition-induced alterations in gut microbiota may increase the sensitivity to anxiety-inducing gastrointestinal hormones released during meals, one of which is cholecystokinin (CCK). The experimental injection of a high dose of its CCK-4 fragment in healthy individuals induces panic attacks, probably via the stimulation of CCK receptors in the brain. Such meal-related anxiety attacks may take part in developing the clinical picture of AN. Malnutrition may also cause increased effects from appetite-reducing hormones that also seem to have roles in AN development and maintenance. The scientific background, including clinical, microbiological, and biochemical factors, of AN is discussed. A novel model for AN development and maintenance in accordance with this hypothesis is presented. Suggestions for future research are also provided.

## 1. Case Vignette

Jenny, aged 15 years, was taken to her physician by her parents due to weight loss. She was somewhat reluctant to come, but after a while, she explained that she had decided to improve her health, eat less chocolate, and lose some weight 6 months previously. This was initially rather difficult because she was very hungry, but hunger was not a problem after some weeks. She lost some weight, and her friends were impressed by her strong personality. However, there were increasing conflicts in the family; Jenny became very irritable and was quarreling with her parents during the meals each day. She spent more time in her room and less time with her friends. After a while, she had anxiety attacks after meals and felt that she should try to have more control of her situation. She was sad about all the quarreling in her family and sometimes felt miserable. Her only experience of success was her control of food intake. Her parents told the physician that she had lost weight continuously to reach a body mass index (BMI) of 14 kg/m^2^. How would you help Jenny and her family?

## 2. Anorexia Nervosa

The onset of anorexia nervosa (AN) is often during early adolescence [1,2,3]. Many patients report that weight loss was initiated by voluntary dieting, but some patients seem to start losing weight due to depression or excessive exercise and others lose weight after trauma episodes such as sexual harassment. Others seem to start their weight loss due to a gastrointestinal (GI) disorder, protracted infection, or other diseases [3]. Many patients have reported reduced appetite shortly after starting weight loss, while others have continuous feelings of hunger [4]. Families often report conflicts in the family, irritability, and that patients seem to be intensely preoccupied with food.

In the landmark study by Keys et al. [5], a group of healthy young males volunteered to be exposed to a 20% loss of body weight. They demonstrated significant social isolation during malnutrition and became irritable and intensely preoccupied with food. Many of them developed rituals related to food preparation and ingestion. When they were allowed to renourish, they re-established normal social activity and all their hunger-related symptoms disappeared [5].

During the initial stages of AN, the patients and their families often report that the patient is irritable, becomes socially isolated and intensely preoccupied with food, and very often develops strange rituals related to food and eating. These symptoms may be regarded as being induced by starvation. During the first stages of the disease, these starvation symptoms usually dominate the clinical picture [6].

However, after a while, the patients usually start to experience anxiety during and after meals [7,8]. At this stage, the patients often express a wish to control their situation. Unfortunately, the attempts to take control usually imply taking control of food intake and other weight-regulating activities such as excessive exercise [7,9]. Some patients start to vomit after meals and may report relief after having emptied their stomachs. However, the attempts to take control usually cause further reductions in food intake. The response to any stressful experience is usually a further reduction in food intake causing event- and mood-triggered changes in eating [7]. This is usually part of the classical picture of the overvaluation of shape and weight, as well as extreme dietary rules. Many patients report that it is difficult to concentrate on anything other than shape and weight [7]. In adults and adolescents who have had low weight for a while, the maintaining psychological symptoms are prominent and will usually dominate the clinical presentation together with underweight. Both the social and medical consequences of the malnutrition affect the life of a patient.

Unfortunately, the time from AN onset to treatment initiation is often unreasonably long [2], with one study finding that there was a mean of 30 months in adults and adolescents from the debut of symptoms to the start of treatment [2]. Although early intervention should have priority, few therapists have extensive experience with the early stages of AN [10,11]. However, based on the existing literature and clinical experiences, a model for AN development and maintenance is described in Figure 1.

When AN is developed, it can be diagnosed according to The Diagnostic and Statistical Manual of Mental Disorders Fifth Edition [12] (Table 1). Diagnosis is usually straightforward, but additional information from parents may be necessary for young patients. A subpopulation of patients presenting with low body weight have food avoidance or restriction as core symptoms without a significant fear of food. These patients may be diagnosed with aversive/restrictive food-intake disorders [13]. Several somatic disorders can mimic AN, and a differential diagnostic evaluation might be indicated [2,3].

### 2.1. Treatment

The mainstay of treatment is outpatient psychotherapy, but inpatient treatment is sometimes indicated [2,3]. Several different psychotherapeutic approaches are used. For children and adolescents, family-based treatment (FBT), as described by Lock, is considered to have the strongest evidence. The guidelines from the National Institute for Health and Care Excellence in Great Britain recommend the following three treatments (which have little or no difference in effect for adults): cognitive behavior therapy with an eating-disorder focus, Maudsley Model of Anorexia Nervosa Treatment for Adults, and Specialist Supportive Clinical Management [2,15].

### 2.2. Severe and Enduring Disease

The outcomes of AN are unacceptably poor. When patients develop the persistent psychological mechanisms typical of AN in adults, treatment is very difficult. The patients spend most of their time thinking of food and about their body size and shape. After some time, the clinical picture is dominated by the overvaluation of shape and weight, intense fear of weight gain, restraints (attempts to reduce food intake), and restriction (food-intake reduction). Everyday ordinary stressful experiences often cause further reductions in food intake (Figure 1). Although patients might have reached the conclusion that weight gain could be helpful, the persisting psychological mechanisms block the possibility of reducing malnutrition and patients often need specialist healthcare in order to address these mechanisms and gain weight.

No pharmacological treatments have been shown to be effective in treating AN [1,16]. If a patient is able to engage in therapy and their malnutrition is reduced, postmeal anxiety and fear of weight gain usually gradually subside [17]. The vicious cycle described in Figure 1 is gradually broken down if treatment is successful. However, many patients are unable to complete treatment, and even with the best psychotherapies, the dropout rate among adults is 20–50% [18]. In addition, the relapse rates vary between 10% and 70% after completing therapy, with the rates being highest among adult patients. BMI at the end of treatment seems to be a dominant predictor of the risk of relapse, and a BMI < 20 kg/m^2^ at the end of treatment seems to be associated with a high risk of relapse [19].

A significant subset of patients (around one in five) experience recurrent treatment failure and relapse after treatment. If AN has persisted for more than 3 years and two qualified treatments have failed, a patient is considered to have severe and enduring AN (SE-AN). Social isolation develops as a consequence of the weight loss, and a loss of contact with friends and family is very prominent. Mortality in SE-AN remains unacceptably high [2]. SE-AN is often described as the most important healthcare issue to address for AN patients [20].

### 2.3. Epidemiology

Reportedly 92% of individuals affected by AN are female [21], but all sexes, sexual orientations, and ethnicities are affected [22]. The most common age at onset is 15–25 years. The incidence is low in children aged 4–11 years but significantly increases in those older than 11 years. Onset is rare after the age of 30 years [3,23,24].

The estimated prevalence of AN among young females is 0.3%, and it affects up to 4% of females and 0.2% of males during their lifetime [25]. The risk of developing AN seems to be related to social pressure towards being underweight. Beauty models and athletes in sports who receive benefits from being underweight are examples of activities and professions with a high AN risk [3].

### 2.4. Genome-Wide Association Studies on Anorexia Nervosa

Psychiatric genomics has led to unprecedented advances in the understanding of the biology of mental illnesses, including AN [26]. A genome-wide association study (GWAS) [27] identified significantly positive genetic correlations of AN with obsessive compulsive and major depressive disorders. Genes associated with insulin resistance, fasting insulin, leptin, body fat percentage, fat mass, and BMI have significant negative correlations with AN. These associations remained significant after excluding the effect of BMI, meaning that AN shares genetic variations with these metabolic phenotypes that may be independent of BMI. The authors of that study pointed out that a low BMI has traditionally been viewed as a consequence of the psychological features of AN (the drive for thinness and body dissatisfaction). This perspective has failed to yield interventions that reliably lead to sustained weight gain and psychological recovery [27]. They concluded that fundamental metabolic dysregulation may contribute to the exceptional difficulty experienced by individuals with AN in maintaining a healthy BMI [27].

### 2.5. Predisposing and Precipitating Factors

Anxiety disorder frequently exists before AN and may be a predisposing factor for its development [8,28,29]. In a study by Godart et al., three of four patients with AN had a prior diagnosis of at least one anxiety disorder; among these, social phobia, simple phobia, generalized anxiety disorder, and panic disorder (PD) are the most common anxiety diagnoses that occur prior to AN onset [8].

Depression can also be a predisposing or precipitating factor for AN [28]. Some patients report that the preceding depression often causes reduced food intake. In some patients, trauma such as sexual harassment or sexual assault can cause significant reductions in food intake, with weight loss and an increased risk of developing AN [30]. Infections and GI disorders can predispose one to or precipitate AN [31,32]. During the development of type 1 diabetes, there is often a significant weight loss due to decreased insulin secretion before the diagnosis. Rapid weight gain often occurs after insulin administration [33,34,35]. Some patients start to reduce their insulin dose in order to reduce their weight gain, causing subsequent weight loss. This behavior increases the risk of developing AN [36]. Both psychiatric and somatic diseases can therefore predispose on to and trigger AN. Although patients often report that these conditions precede weight loss, it can sometimes be difficult to determine the exact chronological order of the diseases [8].

### 2.6. Comorbidities and Complications

Both psychiatric and somatic comorbidities and complications are common in AN. The most common psychiatric comorbidities are mood and anxiety disorders [37], obsessive-compulsive disorders, personality disorders, substance-use disorders, and neurodevelopmental disorders such as autism spectrum and attention-deficit hyperactivity disorders [2]. The prevalence of comorbid or complicating mood disorder has been reported to vary between 31% and 88.9% in patients with AN [28]. Comorbid disorders tend to worsen the prognosis of AN because they interfere with the treatment response [38,39,40,41,42]. GI diseases are also common comorbidities and complications of AN [43,44]. Type 1 diabetes with comorbid AN is uncommon, but it can be a challenging comorbidity [35].

## 3. What Is the Core Pathology of Anorexia Nervosa?

Many psychiatric and somatic diseases can be predisposing, precipitating, comorbid, or complicating diseases in AN. However, the core pathology of AN that causes AN symptoms has been difficult to identify. Many theories have been presented to attempt to explain and understand the mechanisms of AN development and maintenance. Studies on cognitive defects and neuroimaging have identified altered cognitive function and altered brain activity, but it has been difficult to discern between complications of AN and its etiology [45,46,47]. AN has traditionally been viewed as a consequence of its psychological features: a drive for thinness and body dissatisfaction. This perspective has failed to yield interventions that reliably lead to sustained weight gain and psychological recovery [27]. Attempts to describe and understand AN through evolutionary theories such as the suppression of reproduction or sexual competition theories, as well as the famine hypothesis [48,49], have also failed to provide a comprehensive understanding of AN development.

### Can Studies on Treatment Help Identify the Core Pathology?

Treatment effects are generally poor. However, the last two decades have brought some more effective psychotherapies for children, adolescents, and adults. The mainstay of treatment is outpatient psychotherapy [3]. However, even when methods with the greatest amount of documentation are used, a significant subpopulation is unable to complete treatment and 20–50% of the patients who start psychotherapy drop out [18]. When a large proportion of a study population drops out of treatment, the risk of bias is significant and it can be difficult to determine what is underlying the treatment effect. In addition, there is often a selection bias among patients entering treatment studies. The therapies with the strongest documented effects have very different approaches [7,50,51,52]. The two psychotherapies with the best documentation for effect on AN (FBT) and cognitive behavior therapy enhanced for eating disorders (CBT-E) are both agnostic about AN etiology [51]. Despite several differences, the general strategy that is consistent between FBT and CBT-E is in addressing the persisting mechanism of the eating-disorder psychopathology, especially undereating, rather than exploring any potential causes of the psychopathology [51].

A treatment that focuses only on how to learn to eat sufficiently (the Mandometer treatment) has had good results reported in treating AN, with low relapse rates. In this treatment, the patients receive feedback on how to eat but are not offered psychotherapy [53]. Many patients diagnosed with AN have been treated using this method, with many being referred after AN was diagnosed made by an external physician [54]. It has been compared with other treatments [55].

Thus, psychotherapy focusing on persisting psychological mechanisms seems to be essential for most treatments, but the only common factor for all therapies with documented effects seems to be the intense and effective focus on refeeding to establish a healthy body weight [3,15,52,53,56].

During successful treatment, postmeal anxiety gradually decreases in parallel with a reduction in malnutrition [17]. Patients who are able to complete treatment including establishing a BMI >20 kg/m^2^ and addressing persistent psychological mechanisms often enter long-term remission or are cured [3,7,15,56]. Some patients with healthy BMIs report that the persisting psychological symptoms are minimal shortly after end of treatment [3,56], while in some studies, these mechanisms declined or disappeared the first year after the end of treatment [19,57].

Taking the above findings together, the experiences from successful treatments indicate that results are best when addressing the persisting mechanism of the eating-disorder psychopathology (especially undereating), rather than exploring any potential causes of the eating-disorder psychopathology. These observations are crucial for successful treatment but provide little information about the core pathology of AN.

## 4. The Gut Microbiota–Brain Axis

The gut microbiota–brain (GMB) axis is a dynamic bidirectional system that consists of three different pathways: the neuroendocrine, vagus nerve, and immune pathways [58]. Microbiota refers to the total population of microorganisms including bacteria, viruses, archaea, protists, and fungi in several human tissues and biofluids, including the skin, lungs, mucosa, saliva, and GI tract [59].

### 4.1. The Neuroendocrine Pathway

The neuroendocrine pathway includes a large number of mediators able to communicate with the brain via the blood. One of these pathways, the hypothalamic–pituitary–adrenal axis, mediates the neuroendocrine adaptation to stressful situations of organisms [58]. Stressors stimulate this axis with high levels of glucocorticoids (mostly cortisol) to prepare the body for a real or anticipated challenge [58].

### 4.2. The Vagus Nerve Pathway

The easiest way for the gut microbiome to communicate with the brain is via vagus nerve signaling. The vagus nerve comprises 80% afferent and 20% efferent fibers. The afferent fibers innervate the gut to sense chemical or mechanic stimuli and bring the signals to the nucleus tractus solitarius in the medulla. Enteroendocrine cells can secrete neurotransmitters such as glutamate and 5-hydroxytryptamine (5-HT) at epithelial synapses, which directly act on the vagal neurons [58]. The vagus nerve is a part of the autonomic nervous system (ANS) which has the ability to modulate intestinal barrier integrity, GI motility, secretory processes, and the mucosal immune response. These ANS-induced changes in the microbial habitat can affect intestinal transit in certain regions of the gut. These effects are highly variable, including diurnal variations, and have the ability to modulate the microenvironment of the lumen through changes in gut water content and nutrients [60].

### 4.3. Immune Pathways

Foreign invaders may attack the host organism, which activates immune responses and inflammatory reactions. The intestinal barrier contributes to gut homeostasis maintenance and intestinal inflammation avoidance as it coordinates the crosstalk between the gut microbiome and immune system [58]. Microbe-associated molecules activate different immune-system cells, and gut-associated immune cells can produce proinflammatory cytokines that are able to reach the brain by crossing the blood–brain barrier (BBB) via either diffusion or cytokine transporters [60].

## 5. Barriers to Gut Microbiota–Brain Signaling

The two main barriers to GMB signaling are the intestinal barrier and the BBB [61]. Both barriers are dynamic. Gut microbiota, inflammatory signals, stress and a large number of mediators can alter their permeability. In a healthy state, both barriers are tight and prevent microbiota-related immune signaling to the brain [60]. However, substances or signals transferring across these two barriers may play essential roles in the development and maintenance of several disease states [62,63,64].

### 5.1. The Intestinal Barrier

The intestinal barrier consists of two principal layers: (1) epithelial cells connected by tight junctions [65] and (2) a mucus layer that consists of complex sugar molecules (glycans) with varying thicknesses and compositions [66]. The thinning of the outer layer increases the likelihood that cell-wall components of commensal bacteria come into contact with toll-like receptors (TLRs) in the cell wall, which triggers cytokine release. Cytokine release can also lead to the loosening of the tight junctions between epithelial cells, allowing microorganisms or their fragments to transfer across the intestinal barrier [67].

Short-chain fatty acids (SCFAs) are signaling molecules exclusively generated by gut microbes through the fermentation of dietary fiber (complex carbohydrates), since humans lack the enzymes required to digest fiber. Complex carbohydrates are metabolized by intestinal microorganisms into short chain fatty acids (SCFA, n-butyrate, acetate, and propionate) with neuroactive activity. The SCFAs produced by certain gut microbes are crucial in maintaining the integrity of the intestinal structure by maintaining the tight junctions between cells and reducing gut-associated immune cell activation [60]. SCFAs act on a wide range of targets via the activation of free fatty-acid receptors (FFARs) and have been implicated in physiological processes ranging from neuroplasticity to gene expression, food intake, and immune system modulation. SCFAs also stimulate CCK-gene expression [68]. In addition, SCFAs are involved in satiety regulation by regulating hormones such as glucagon-like peptide 1 (GLP-1) and protein YY (PYY) [69]. SCFA butyrate downregulates gene expression in the gut-associated immune system, suggesting that decreased levels of SCFA-producing bacteria contribute to increased inflammation. SCFAs also modulate the synthesis of 5-HT, lipopolysaccharide (LPS), and bacterial lipoprotein—as well as activating different immune system cells such as macrophages, neutrophils, and dendritic cells— probably via TLR activation. TLRs play a major role in the molecular communication between gut microbiome and homeostasis alterations in the immune system [60,70]. SCFAs are also able to cross the BBB and impact neural circuits [71].

In addition, stress induces endotoxemia and low-grade inflammation by increasing barrier permeability [72,73]. Increased intestinal permeability and low-grade inflammation in the colon were observed in an activity-based anorexia rodent model [74].

SCFAs also regulate physiological intestinal functions, including those involving motility, secretion, and inflammation (see below) through their cognate FFARs.

### 5.2. The Blood–Brain Barrier

The vascular BBB comprises specialized brain endothelial cells that restrict the passage of plasma proteins into the central nervous system, act as a regulatory interface between the brain and blood, and play nutritive, homeostatic, and communication roles. Tight-junction proteins are specialized features of the endothelial cells in the brain that restrict the paracellular diffusion of substances between the blood and brain. The disruption of tight junctions can lead to a leaky BBB [75]. Bacteria and their cell-wall constituents can cause BBB dysfunction via several mechanisms [60], and two of these mechanisms will be described here: SCFA-mediated regulation and by immune mechanisms which are also partly SCFA-mediated.

In the SCFA-mediated mechanisms, gut-produced SCFAs are released into the blood. SCFAs can cross the BBB via monocarboxylate transporters located on endothelial cells and influence the integrity of the BBB by upregulating tight-junction protein expression. Bacteria such as *Clostridium tyrobutyricum* (which produces high levels of the SCFA butyrate) have been found to improve the integrity of the BBB in germ-free (GF) mice, which was associated with the upregulation of tight-junction protein expression [75].

In the immune mechanisms, the gut microbiota and its factors can alter peripheral immune cells to promote interactions with the BBB [75]. Bacteria and their factors or cytokines and other immune-active substances released from peripheral sites under the influence of the gut microbiota can cross the BBB, alter its integrity, change BBB transport rates, or induce the release of neuroimmune substances from the barrier cells [75]. The gut-produced SCFAs can activate FFARs, TLRs, and immune cells. When activated, gut-associated immune cells produce proinflammatory cytokines (interleukin (IL)-1α, IL-1β, IL-6, and tumor necrosis factor α) that can reach the brain by crossing the BBB via either diffusion or cytokine transporters [60]. Once they reach the brain, these cytokines can act on receptors on microglia and stimulate further cytokine release and neuronal function modulation. Injecting animals with LPS was shown to greatly increase BBB permeability, which indicates that disrupting the BBB can be caused by systemic immune activation [60].

## 6. The Gut Microbiota–Brain Axis and Anorexia Nervosa

The intestinal microbiota seem to plays a crucial role in metabolic function, immunomodulation, and weight regulation in AN [59,76,77,78,79,80,81,82,83].

At birth, the gut is mainly colonized with bacteria from the mother, and the environment dominates over host genetics in shaping human gut microbiota [84]. Diet shapes the composition of gut microbiota [59], and dietary changes may easily provoke intense gut microbiota shifts [76,85].

Studies on patients with AN revealed that several species were significantly reduced [86]. Di Lodovico et al. systematically reviewed previous studies evaluating gut microbiota in anorexia nervosa (AN) and healthy individuals by performing a quantitative synthesis of the pooled clinical and microbiological data. They found an increase in intestinal mucin-degrading bacteria and a decrease in the short chain fatty acid (SCFA) butyrate microbial producers in patients with AN. Increased mucin-degraders and decreased butyrate indicate possible decreased intestinal barrier function [87].

Mason et al. characterized gut microbiota in patients with AN and found increased populations of *Methanobacterbrevi smithii*, which has an important role in the efficiency of microbial fermentation and the associated energy yield from one’s diet [88].

Borgo et al. also found increased levels of *Methanobacterbrevi smithii* and decreased levels of the butyrate producers *Roseburia* and *Clostridium*. They also observed an increase in *Enterobacteriaceae*, which is normally associated with intestinal inflammation and can produce the bacterial peptide caseinolytic protease B (CLPB), which has an inhibitory effect on appetite [89].

The disruption of the composition of the intestinal microbiota has been observed in various diseases, including AN [90]. Longitudinal studies of patients with AN found that gut bacterial diversity was significantly decreased at the start of treatment and significantly increased at discharge [91]. The relative abundance and diversity of bacterial taxa in the gut are regarded as disease-severity markers in AN [16].

The BBB was shown to be more permeable in GF mice than in control mice, and introducing healthy microbiota into the mice partially restored their barrier function [92]. The colonization of the gut with SCFA-producing bacteria also decreases BBB permeability, suggesting that SCFAs play important roles in BBB development and maintenance [60].

In patients with AN, a decrease in intestinal permeability based on urinary lactulose recovery has been reported in the small intestine [93]. Whether intestinal permeability is increased in the colon of patients with AN remains unknown [94]. One study found decreased levels of the intestinal permeability marker zonulin in patients with AN compared to healthy controls. In the patients with AN, BMI significantly predicted serum zonulin levels [95].

Taking the above findings together, several studies have indicated that the composition of gut bacteria and SCFA release is altered in a way that may increase intestinal and BBB permeability in AN.

### 6.1. The Neuroendocrine Interaction in the Gut Microbiota–Brain Axis in Anorexia Nervosa

Enteroendocrine cells of the gut play essential roles in sensing nutrient supplements in the gut and regulating the release of gut peptides, mostly cholecystokinin (CCK), GLP-1, and PYY. These cells are essential in controlling the gut-to-brain signaling that regulates food intake and energy expenditure [58,96].

During a meal, many hormones are released to regulate digestion, some of which exert anxiogenic effects under certain conditions. The GI hormone CCK is mostly synthesized in the proximal small intestinal endothelium and the brain [97]. Only one CCK mRNA molecule has been found; it codes for a pro-CCK protein and many CCK peptides, which are fragments of CCK and are all ligands for CCK-1 and CCK-2 receptors. The CCK-1 receptor mediates gall bladder contraction and pancreatic enzyme secretion, as well as delaying gastric emptying. The CCK-2 receptor is also referred to as the brain receptor. This receptor is less specific than the CCK-1 receptor and binds to various CCK fragments. Fat- and protein-rich foods are the most important stimuli for CCK release. Within 20 min of consuming a meal, the concentration of CCK in plasma increases by 3–5 times and then gradually decreases until it peaks again after 1.5 to 2 h. In addition to inducing gallbladder contraction and pancreatic enzyme release, the hormone also induces satiety and regulates gut motility by interacting with the CCK-1 receptor [97].

Injecting a high dose of the CCK fragment CCK-4 into humans induces panic attacks, and it has also been found to induce anticipatory anxiety even in healthy individuals [98,99]. The subjective panic response has been assessed using the Panic Symptom Scale and functional magnetic resonance imaging (MRI) [97,98]. There is considerable evidence for a role of the neuropeptide CCK in PD. This effect has been attributed to increased sensitivity to CCK fragments in the brain’s CCK-2 receptor [100,101]. Positron emission tomography (PET) of healthy humans experiencing CCK-4-induced panic attacks has indicated that the brainstem is involved in the initial reaction to CCK-4 [102]. Some regions of the brainstem have little or no BBB protection. One of these, the area postrema, triggers nausea and vomiting [103].

Cuntz et al. assessed the levels of CCK-like reactivity in patients with AN during treatment [104]. The CCK levels in patients with AN were similar to those in controls both prior to and after a test meal. Pre- and postmeal CCK levels significantly increased after an initial weight gain but decreased again with further weight loss. There was no significant association between CCK release and either initial weight, BMI, or changes therein. These observations indicate that CCK-4 levels are not affected by malnutrition. Although CCK-4 sensitivity has been investigated in patients with major depression, irritable bowel syndrome, and premenstrual dysphoric disorder [105,106,107], there are no data available on CCK-4 sensitivity in AN.

Present knowledge indicates that malnutrition-induced alterations in gut microbiota may induce increased BBB permeability. This conclusion is mainly based on animal studies [60,75]. However, some studies in humans have also indicated that gut microbiota may affect BBB function [108]. The panicogenic substances released during meals may then reach their receptors in the brain and induce anxiety when a susceptible malnourished individual starts eating. In addition, as PET studies in healthy individuals indicate that CCK-4 may exert its initial effect in the brainstem areas where there is little or no BBB function [102], the brainstem ANS nuclei might be involved in the precipitation of the anxiety reactions.

Anxiety reactions induced by CCK-4 may, in turn, induce anticipatory anxiety, the overvaluation of shape and weight, and problems with the correct assessment of body shape and weight by a mechanism described for anxiety disorders [7,29].

### 6.2. Appetite Regulation in Anorexia Nervosa

Food intake in AN seems to be regulated by two main mechanisms: (1) anxiety attacks and maintaining psychological mechanisms that (according to the current hypothesis) are induced by food intake in malnourished patients and (2) appetite regulators. The loss of or significantly reduced appetite is often reported to occur after some weeks of dieting. However, the degree of loss of hunger varies, and many patients experience persisting and sometimes intense hunger [6].

The GMB axis seems to play an essential role in appetite regulation. Several GI hormones are secreted by enteroendocrine cells in the gut and pancreas, including ghrelin, insulin, GLP-1, PYY, pancreatic polypeptide, secretin, glucose-dependent insulinotropic peptide, and CCK. They have major impacts on the energy balance and homeostasis maintenance by inducing satiety, and meal termination is mostly affected by appetite-regulating centers in the brain [109].

A recent systematic review of animal models for AN found that one model (of activity-based AN) supported that injecting orexigenic ghrelin reduced the risk of AN-development, which suggests that appetite stimulation may acutely affect the risk of AN [110,111].

*E. coli*, which mainly inhabits the lower intestinal tract in humans, releases a protein that secretes the caseinolytic peptidase B (CLPB) in response to oxidative stress and probably other stress-related events [88]. The CLPB protein has a anorexigenic effect that is probably related to two mechanisms. CLPB seems to increase PYY secretion. PYY is released from the intestine mucosa in response to food. The plasma levels of PYY start to rise within 15 min after starting to eat and plateau within 90 min, remaining high for up to six hours. The main effect of this is the inhibition of food intake. [112,113,114]. Plasma PYY levels are elevated in AN [115], and animal experiments have indicated that PYY also is involved in emotional–affective behavior [115].

CLPB also induce autoantibodies that cross-react with anorexigenic alpha melanocyte-stimulating hormone (α-MSH) and interfere with appetite regulation by affecting hypothalamic energetic-homeostasis-regulating centers [114] by stimulating anorexigenic neurons [89]. CLPB and antibodies against α-MSH have been found in patients with eating disorders [88,116].

In summary, appetite regulation is complex and appetite is significantly decreased in some patients with AN, especially during its initial stages. Malnutrition-induced alterations in the GMB axis might decrease hunger signals in these patients. The reduction in hunger during the initial stages of the disease may play a role in AN development. Many patients report persistently reduced appetite that may influence the progress of psychotherapy. However, some patients experience intense hunger but are still unable to increase their food intake due to intense, persisting psychological mechanisms [7].

## 7. Hypothesis: Malnutrition-Induced Alterations in the Gut Microbiota–Brain Axis Are the Core Pathologies in Anorexia Nervosa

The proposed hypothesis is based on the observation that the initial weight loss, which seems to be the first step in AN development, is often induced by voluntary dieting but also be caused by psychiatric or somatic diseases (Figure 1). During this weight loss, appetite regulation might be disturbed by an increased release of gut bacterial fragments with the capacity to reduce food intake and a malnutrition-induced decrease in bacterial SCFA release. Malnutrition- or exercise-induced immunological reactions may also be part of an appetite regulation problem [74]. Malnutrition-induced increased sensitivity to CCK-4 may induce anxiety reactions during meals. As a consequence, patients develop anticipatory anxiety, the overvaluation of their body shape and weight, and misconceptions about their body shape and size (Figure 2).

## 8. Discussion

Significant amounts of information are available about the epidemiology, mortality rates, predisposition, precipitation, comorbidities, and complicating diseases of AN. We know that establishing a healthy BMI is necessary for successful recovery with low relapse rates. Although psychotherapies have produced acceptable effects in some studies performed over the past 20 years, the dropout rates, relapse rates, and prevalence of SE-AN remain unacceptably high [20]. Numerous pharmacological interventions have been tested in AN, and medications that often have beneficial effects in various psychiatric diseases have not been found to have any effect on AN [1]. The prevailing theories for the cause of the weight loss and psychological symptoms have not led to interventions that reliably sustain weight gain and psychological recovery [27]. There is still no generally accepted, comprehensive theory that explains the core pathology of AN.

GWASs indicate that anxiety and depression disorders may predispose one to AN development. Susceptibility to certain metabolic disturbances, including increased insulin sensitivity independent of BMI, also seems to predispose one to AN development. Malnutrition has a well-known and clinically relevant enhancing effect on insulin sensitivity, but the findings of GWASs indicate that patients with AN may be especially susceptible to developing increased insulin sensitivity. The findings of studies prompted suggestions that AN should be reconceptualized as a metabo-psychiatric disorder. It is therefore relevant to hypothesize that increased sensitivity to other metabolic hormones plays a role in AN development and maintenance.

Many studies have suggested that intestinal microbiota play a crucial role in metabolic function, immunomodulation, and weight regulation in AN [76,77,78,79,80,81,82,83]. However, little is known about the exact interaction mechanisms in AN development and maintenance. This review presents a hypothesis aimed at describing how malnutrition-related effects on GMB signal transduction may represent the core pathologies of AN and may explain the AN symptoms. The strength of this model is that it can explain how voluntary or disease-induced malnutrition may cause gut microbiota alterations that may induce AN symptoms. Malnutrition-induced alteration in the gut microbiota and SCFAs may increase BBB permeability, therefore allowing panicogenic hormones to reach their receptors in the brain and induce anxiety reactions during meals, in turn inducing anticipatory anxiety. The anxiety reactions and anticipatory anxiety may then induce problems related to self-assessments of body shape and size. Similar mechanisms have been described for other anxiety reactions such as arachnophobia, in which the patient tends to overestimate the size of a spider [7,29].

Increased sensitivity to CCK-4 may also be caused by other mechanisms than increased BBB permeability. PET studies on healthy humans exposed to CCK-4-induced panic attacks have indicated that the brainstem seems to be involved in the initial stage of these panic attacks in individuals with a healthy weight [102]. These observations suggest that anxiety reactions can be initiated by brainstem-regulated mechanisms [103].

Research on PD without AN has suggested that the panic response among PD patients may be related to the increased sensitivity of the brain’s CCK-2 receptor [101]. Thus, PD patients will probably respond to physiological concentrations of CCK-4. A third potential mechanism for increased CCK-4 sensitivity in AN may be malnutrition-induced alterations in the CCK-2 receptor.

Appetite regulation in AN seems to be complex. Patients with AN often report a loss of appetite during the early stages of the disease. If young patients experience unpleasant reactions to food, they might express their unwillingness to experience these reactions by not being hungry. However, there is probably a significant reduction in appetite in some patients, perhaps most clearly expressed during the first stages of the disease. However, other patients report that they have persisting intense hunger [6].

Malnutrition of gut microbiota seems to cause the increased passage of the CLPB fragments of *E. coli* from the gut to the blood [114] (Figure 2). CLPB may cause the increased released of PYY from the intestine mucosa in response to food. The main effect is the inhibition of food intake, and PYY has been found to be elevated in AN [115]. The plasma levels of PYY start to rise within 15 min after starting to eat, plateau within 90 min and remain high for up to six hours, thus indicating that the PYY inhibition of appetite may last long enough to significantly reduce appetite for the next meal and thus exert a persisting inhibition of food intake [115]. In addition, animal experiments indicate that PYY is also involved in emotional–affective behavior [115]. These observations suggest that PYY may be part of the food-induced negative experiences in patients with AN.

Further, various immunological reactions induced by malnutrition and stress may also enhance appetite loss. The hypothesis describing the malnutrition-induced alteration of the GMB axis as the core pathologies of AN may also be relevant for the mechanisms of successful AN treatment. If a patient is able to reach a healthy weight, their gut bacteria may have sufficient nutrients for proliferation, and a normal balance between bacterial strains can be re-established if all important bacterial strains have survived the malnutrition. The dysbiosis-related persisting mechanisms that cause the passage of panicogenic substances to their receptors in the brain and postmeal anxiety will gradually decrease and eventually disappear [17].

However, in order to obtain long term remission, both a normal BMI and capacity to effectively address psychological mechanisms are essential [19,117]. If the maintaining psychological mechanisms are still operating, they might be reactivated by external triggers such as social pressure towards dieting. Then, the malnutrition-induced effects on GMB axis may recur and patients could relapse. If the gut microbiota at the end of treatment still have significant dysbiosis, this may also induce altered GMB axis function with risk of relapse.

There are several limitations related to the present hypothesis. Firstly, several weeks or even months can often pass before patients and their parents discover that the patient might have an eating disorder. Many patients also consider the weight loss during the initial stage of the disease to be beneficial. If the weight loss is voluntary, the patient often receives positive responses from friends and even trainers and teachers. Irritability and isolation might be regarded as natural parts of normal development in early adolescence. Nearly all information about the first months of the disease are therefore based on retrospective information collected from patients and their relatives. Given that the mean time between symptom onset and the start of treatment is 30 months, it is often difficult to obtain a detailed description of the first months of the disease [2,50].

Secondly, much of our knowledge about the GMB axis is based on animal studies. Information obtained from mice studies might not be relevant to humans, and the observations are especially not freely transferable to patients with AN. The metabolism of AN patients often significantly differs from that in animals, making the latter less relevant for healthy humans. Although the present hypothesis is compatible with the clinical observations and our understanding of epidemiology and genetics in AN, it still remains to be tested.

A third limitation is that around 10–34% of the patients with AN have PD before they develop AN [8]. Patients with preexisting PD will probably have increased CCK-4 sensitivity, but we do not know whether increased CCK-4 sensitivity is present in other patients with AN. The proposed model suggests that all patients with AN may have malnutrition-induced increased sensitivity to CCK-4.

The clinical picture at presentation with AN will vary depending on age of the patient, preceding and precipitating disorders, and comorbid and complicating diseases. Anxiety reactions or other reactions related to food intake may not be detectable, and the DSM-5 criteria of AN are not sometimes prominent. Defining diagnostic groups to assess the relevance of the proposed model might be challenging.

Epidemiological data and genome-wide association studies indicate that anxiety and depressive disorders may predispose one to the development of AN. In addition, the GWAS studies indicate that metabolic factors play an essential role in AN. The crucial role of BMI in development, maintenance, and relapse suggest that a somatic factor may play an essential role the pathogenesis of the disease. Data from a large number of studies indicate that the GMB axis is involved in the maintenance of the disease and seems to be the leading candidate as the mediator of the importance of BMI in the pathogenesis [76,77,78,79,80,81,82]. Based on our present knowledge, it is reasonable to hypothesize that GMB axis disturbances are the core pathologies of AN and that malnutrition-induced alterations in gut microbiota can be an early step in the pathogenesis of AN. However, the interactions within the gut microbiota and in the GMB axis in humans seem to be extremely complex, and it will probably be challenging to test these hypotheses. However, clinical data indicate that new interventions and new models that reliably lead to sustained weight gain and psychological recovery are needed [27,118]. Although dropout from treatment, rates of relapse, and SE-AN are unacceptably high, the results from some treatments indicate that a lasting cure is possible for many patients with AN [3]. Knowledge of the effects of interventions aiming at re-establishing normal regulation of feeding behavior in patients with AN is essential. The GMB axis seems to be an important regulator in AN and the malnutrition-induced alterations in the GMB axis may be the core pathologies in AN.

## 9. Future Research

Studies on interventions aiming at re-establish normal signal transduction in the GMB axis are under way and will probably shed more light on our understanding of the core pathology of AN [58,76]. Studies on the effects of prebiotics, probiotics and fecal transplantation are mainly aiming at re-establishing normal gut microbiota and thereby re-establishing normal signal transduction in the GMB axis. The use of prebiotics (beneficial live microorganisms) has been studied in related disorders such as anxiety and depression. Some probiotics seem to have significant beneficial effects on anxiety disorders, and the effect of probiotics in reducing anxiety seems to be larger the higher the baseline anxiety level of the individual is [76]. A meta-analysis with a total of 1349 patients showed that probiotic administration had positive and significant effect on mood, especially on those with mild-to-moderate depressive symptoms [58,119,120]. The first clinical trial evaluating the effects of probiotics in treating AN has been started, though the outcomes have yet to be published [58,121]. In addition, hypotheses such as the one presented in this review should be carefully tested by the injection of the candidate regulator, such as CCK-4. Imaging, including PET scans, illustrating the effects of injection of candidate regulators should be performed in both cross-sectional and longitudinal treatment studies. In addition, BBB function should be studied in longitudinal treatment studies, e.g., by contrast-enhanced dynamic MRI scanning for the quantitative assessment of BBB leakage in AN [122]. Several other methods are relevant and may aid understanding and address the core pathologies of AN.

## Figures and Tables

**Figure 1 microorganisms-10-01486-f001:**
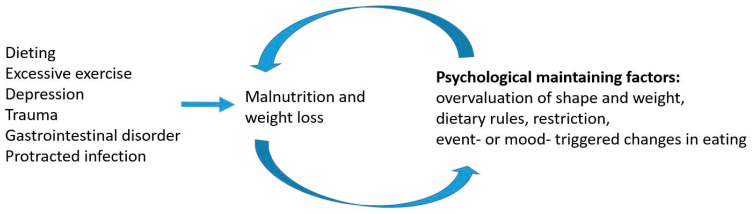
Proposed model describing how AN develops and is maintained in a majority of patients [3].

**Figure 2 microorganisms-10-01486-f002:**
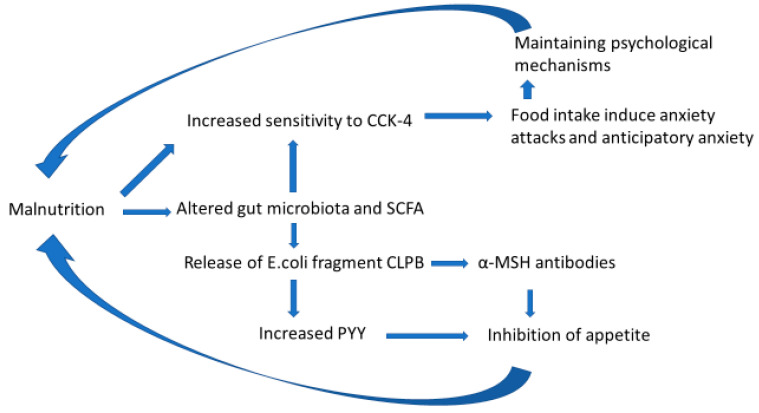
Model illustrating some potential malnutrition-induced neuroendocrine interactions in AN. Malnutrition is proposed to cause increased sensitivity to panicogenic CCK-4. This may be caused by malnutrition-induced alterations of gut bacteria with concomitant SCFA alteration and increased BBB permeability or other mechanisms. Food intake will induce CCK-4 release, and the brain’s CCK-2 receptor will induce anxiety attacks due to increased CCK-4 sensitivity. These attacks may induce anticipatory anxiety, and the development of maintaining psychological mechanisms with further weight loss. Malnutrition also seems to be associated with the release of the fragment CLPB from gut *E. coli* to blood. The CLPB fragment induces α-MSH antibodies and increased PYY that, in turn, inhibit food intake. AN: anorexia nervosa; CCK-4: fragment of cholecystokinin; *E. coli*: *Escherichia coli*; CLPB: caseinolytic peptidase B; α-MSH: alpha-melanocyte-stimulating hormone; PYY: protein YY.

**Table 1 microorganisms-10-01486-t001:** Diagnostic criteria, subtypes and severity of anorexia nervosa.

**Diagnostic Criteria**
Restriction of energy intake relative to requirements in anorexia nervosa leads to a significantly low body weight for the patient’s age, sex, developmental trajectory and physical health. A significantly low weight is defined as a weight that is less than the minimal normal weight or (in children and adolescents) less than the minimum expected weight. Intense fear of gaining weight or becoming fat, or persistent behaviour that interferes with weight gain, even though the patient has a significantly low weight. Disturbance in the way in which one’s body weight or shape is experienced, undue influence of body weight or shape on self-evaluation, or persistent lack of recognition of the seriousness of the current low body weight.
**Subtype Designation**
Restricting subtype: During the past 3 months, the patient has not engagd in recurrent episodes of binge-eating or purging behaviour (i.e., self-induced vomiting or the misuse of laxatives, diuretics or enemas). Weight loss is primarily caused through dieting, fasting, excessive exercise, or all of these methods. Binge-eating/purging subtype: During the past 3 months, the patient has engaged in recurrent episodes of binge-eating or purging behavior (i.e., self-induced vomiting or the misuse of laxatives, diuretics or enemas).
**Current Severity**
Mildly severe low body weight is defined as a BMI ≥ 17.00 kg/m^2^ Moderately severe low body weight is defined as a BMI of 16.00–16.99 kg/m^2^ Severe low body weight is defined as a BMI of 15.00–15.99 kg/m^2^ Extremely severe low body weight is defined as a BMI < 15.00 kg/m^2^

All three diagnostic criteria are required for the diagnosis anorexia nervosa; BMI: body mass index [1,14].

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
