# Peer review of "Are the Effects of Malnutrition on the Gut Microbiota–Brain Axis the Core Pathologies of Anorexia Nervosa?"

_microorganisms, 2022, doi:10.3390/microorganisms10081486_

Round 1

Reviewer 1 Report

General comment:

The review by Stein Frostad, “Are the effects of malnutrition on the gut microbiota-brain axis the core pathologies of anorexia nervosa?”, is highly topical and interesting. The review includes a hypothesis according to which the gut-brain peptides cholecystokinin (CCK) and peptide YY (PYY) are central pathophysiological players. Since anorexia nervosa is such a serious disease and is so widespread among girls and younger women; since its pathophysiology remains enigmatic and obscure; and – consequently – since therapies of today have so many shortcomings, new suggestions about the pathogenesis and disease-mechanisms are desirable. Therefore, the present review definitely deserves publication – irrespective of attitudes to the described hypothesis.

 Minor comments:

1.       Line 13: The word “dose” (after a high …) is missing.

2.       Line 60: “stomach” with an ‘h’, not a ‘k’.

3.       Lines 310-312: SCFAs also stimulate CCK-gene expression (see Zang et al. “Fructose malabsorption induces cholecystokinin expression in the ileum and cecum by changing microbiota composition and metabolism”. FASEB J 2019;33:7126-7142).

4.       Lines 399-400: It is true that most CCK in the body is expressed in the brain and the gut. But in the gut, the duodenal contribution is considerably smaller than that of the jejunum (because the duodenum constitutes such a short part of the small intestine). Therefore, in this sentence “duodenal” should be replaced by “proximal small intestinal” endothelium. Also, ref. 96 does not state that most CCK originates from the duodenal endothelium.

Author Response

Response to peer-reviewer 1:

General comment:

The review by Stein Frostad, “Are the effects of malnutrition on the gut microbiota-brain axis the core pathologies of anorexia nervosa?”, is highly topical and interesting. The review includes a hypothesis according to which the gut-brain peptides cholecystokinin (CCK) and peptide YY (PYY) are central pathophysiological players. Since anorexia nervosa is such a serious disease and is so widespread among girls and younger women; since its pathophysiology remains enigmatic and obscure; and – consequently – since therapies of today have so many shortcomings, new suggestions about the pathogenesis and disease-mechanisms are desirable. Therefore, the present review definitely deserves publication – irrespective of attitudes to the described hypothesis.

 Minor comments:

  1. Line 13: The word “dose” (after a high …) is missing.
  2. Line 60: “stomach” with an ‘h’, not a ‘k’.
  3. Lines 310-312: SCFAs also stimulate CCK-gene expression (see Zang et al. “Fructose malabsorption induces cholecystokinin expression in the ileum and cecum by changing microbiota composition and metabolism”. FASEB J 2019;33:7126-7142).
  4. Lines 399-400: It is true that most CCK in the body is expressed in the brain and the gut. But in the gut, the duodenal contribution is considerably smaller than that of the jejunum (because the duodenum constitutes such a short part of the small intestine). Therefore, in this sentence “duodenal” should be replaced by “proximal small intestinal” endothelium. Also, ref. 96 does not state that most CCK originates from the duodenal endothelium.

Thank You for Your comments,

Comment 1: I have added “dose” (after a high…)

Comment 2: I have written “stomach” with an ‘h’, not a ‘k’.

Comment 3: Thank You for this most interesting comment. A line stating that SCFAs stimulate CCK-gene expression and the corresponding reference has been added.

Comment 4: Thank You for alerting me to this mistake. The sentence “duodenal” has been replaced by “proximal small intestinal”.

Stein Frostad

Reviewer 2 Report

The work presents a very interesting approach to several fields of study. It is very well written and referenced. However, I suggest that the insertion of an item related to the use of probiotics would be extremely important, considering that this subject has been widely discussed and used for the treatment of these types of diseases. In addition, there are several researchers demonstrating beneficial effects in the use of probiotics in this field, and it is interesting to cite some studies.

Author Response

Response to peer-reviewer 2:

Comments and Suggestions for Authors

The work presents a very interesting approach to several fields of study. It is very well written and referenced. However, I suggest that the insertion of an item related to the use of probiotics would be extremely important, considering that this subject has been widely discussed and used for the treatment of these types of diseases. In addition, there are several researchers demonstrating beneficial effects in the use of probiotics in this field, and it is interesting to cite some studies.

Thank You for the comments and most relevant suggestions for improvement of the manuscript.

An item on probiotics has been added in section 9. Future research.